# Ultrafast creation of large Schrödinger cat states of an atom

K.G. Johnson [1], J.D. Wong-Campos[1], B. Neyenhuis[1], J. Mizrahi[1] & C. Monroe[1]

Mesoscopic quantum superpositions, or Schrödinger cat states, are widely studied for fundamental investigations of quantum measurement and decoherence as well as applications in sensing and quantum information science. The generation and maintenance of such states relies upon a balance between efficient external coherent control of the system and sufficient isolation from the environment. Here we create a variety of cat states of a single trapped atom's motion in a harmonic oscillator using ultrafast laser pulses. These pulses produce high fidelity impulsive forces that separate the atom into widely separated positions, without restrictions that typically limit the speed of the interaction or the size and complexity of the resulting motional superposition. This allows us to quickly generate and measure cat states larger than previously achieved in a harmonic oscillator, and create complex multi-component superposition states in atoms.

[1] Department of Physics, Joint Quantum Institute and Joint Center for Quantum Information and Computer Science, University of Maryland, College Park, MD 20742, USA. Correspondence and requests for materials should be addressed to K.G.J. (email: kale@umd.edu)

Quantum superposition is the primary conceptual departure of quantum mechanics from classical physics, giving rise to fundamentally probabilistic measurements[1], non-local correlations in spacetime[2], and the ability to process information in ways that are impossible using classical means[3]. Quantum superpositions of widely separated but localised states, sometimes called Schrödinger cat states[4], exacerbate the quantum/classical divide. These states can be created in systems such as cold atoms and ions[5–9], microwave cavity QED with Rydberg atoms[10] and superconducting circuits[11–13], nanomechanical oscillators[14], van der Waals clusters and biomolecules[15, 16]. All these systems gain sensitivity to outside influences with larger separations[7, 17].

The natural localised quantum state of a harmonic oscillator is its displaced ground state (coherent state) $|\alpha\rangle$[18], which is a Poissonian superposition of oscillator quanta with mean number $|\alpha|^2$. For a mechanical oscillator with mass $m$ and frequency $\omega$, the complex number $\alpha$ characterises the position $\hat{x}$ and momentum $\hat{p}$ operators of the oscillator, with $\mathrm{Re}[\alpha] = \langle\hat{x}\rangle/(2x_0)$ and $\mathrm{Im}[\alpha] = \langle\hat{p}\rangle x_0/\hbar$, where $x_0 = \sqrt{\hbar/(2m\omega)}$ is the zero-point width; and as energy oscillates between its kinetic $(\hat{p}^2/2m)$ and potential $(m\omega^2\hat{x}^2/2)$ forms, the coherent state makes circles in phase space. Schrödinger cat superpositions of coherent states $|\alpha_1\rangle + |\alpha_2\rangle$ of size $\Delta\alpha = |\alpha_1 - \alpha_2| \gg 1$ have been created in the harmonic motion of a massive particle (phonons)[6] and in a single mode electromagnetic field (photons)[19]. In trapped ion systems, coherent states of motional oscillations are split using a qubit derived from internal electronic energy states[6, 8]. For photonic cat states, coherent states in a single mode microwave cavity are split using atoms or superconducting Josephson junctions. Recent experiments have created cat states with more than two components[20] for qubit storage and error protection[13]. In superconducting cavities, the size of the cat state is restricted to a maximum photon number of $|\Delta\alpha|^2 \sim 100$, due to nonlinearity of the self-Kerr and dispersive shift[13]. For trapped ions, cat states have been restricted to a regime where the motion is near or smaller than the wavelength of light providing the dispersive force, or the 'Lamb-Dicke' regime (LDR), which typically restricts phonon numbers $|\Delta\alpha|^2$ also to a few hundred[21] (see Fig. 1 for relationship of this, and other, atom experiments to the LDR). Multicomponent superposition states have not previously been created in the harmonic motion of trapped ions.

Here we use ultrafast laser pulses to create cat states in the motion of a single $^{171}\mathrm{Yb}^+$ ion confined in a harmonic trap with frequency $\omega/2\pi = 1$ MHz[22]. We characterise the coherence of the cat state by interfering the components of the superposition and observing fringes in the atomic populations mapped to the qubit. We achieve the largest phase space separation in any quantum oscillator to date—a superposition with $\Delta\alpha \approx 24$ (259 nm maximum separation compared to a $x_0 = 5.4$ nm spread of each component), or a separation of $\approx 580$ phonons, with 36% fidelity. The ultrafast nature of the cat generation is less restrictive on nonlinearities in the forces on the atom, and allows for very fast state creation with $\Delta\alpha = 0.4$ per laser pulse period (~12 ns). Finally, we demonstrate a method to create three-, four-, six- and eight-component superposition states by timing the laser pulses at particular phases of the harmonic motion in the trap. These tools allow us to create and measure fragile mesoscopic states before they lose coherence.

## Results

### Generating an ultrafast state-dependent force

In these experiments, the ion is confined in a radiofrequency Paul trap[23], with harmonic oscillation frequencies $(\omega_x \equiv \omega, \omega_y, \omega_z)/2\pi = (1.0, 0.8, 0.6)$ MHz. The two hyperfine ground states of $^{171}\mathrm{Yb}^+$

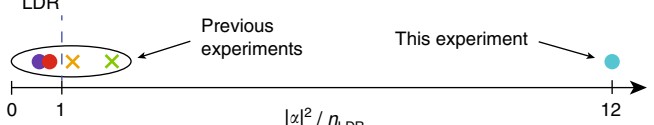

**Fig. 1** Mesoscopic superposition states and the Lamb-Dicke regime (LDR). Mesoscopic superposition states of trapped atom motion have been created in a number of different experimental apparatuses. Here we plot the average phonon number $|\alpha|^2$ from a selection of the largest superposition states created to date, each normalised to the phonon number associated with their LDR boundary $n_{\mathrm{LDR}} = 1/(2\eta^2) - 1/2$. In the experiments denoted by "x" points, the applied force became more strongly position-dependent beyond the LDR, causing significant and unavoidable distortion of the displaced coherent states (*orange*[32], *green*[21]). The circular points signify undistorted displaced states (*red*[6], *blue* (this experiment)) or squeezed states (*purple*[8])

$(|\downarrow\rangle \equiv |F = 0, m_F = 0\rangle$ and $|\uparrow\rangle \equiv |F = 1, m_F = 0\rangle$, with qubit splitting $\omega_{\mathrm{hf}}/2\pi = 12.642815$ GHz) are used to split coherent states of the atom motion through a strong state-dependent kick (SDK)[24]. The qubit can also be coherently manipulated without motional coupling using resonant microwave pulses.

Each experiment follows the same general procedure. We initialise the atom's motion by Doppler laser cooling to an average vibrational occupation number $\bar{n} \sim 10$. This is followed by resolved sideband cooling to $\bar{n} \sim 0.15$. (While we consider the actual thermal vibrational state when comparing data to theory[24], the initial state will be represented from here as $n = 0$ for simplicity.) Optical pumping initialises the qubit state to $|\downarrow\rangle$[25], and then a pair of separated Ramsey microwave $\pi/2$ pulses with variable relative phase is applied to the ion. After the first microwave pulse, the ion is in state $|\psi_1\rangle = (|\uparrow\rangle + |\downarrow\rangle)|n = 0\rangle$ (we suppress normalisation factors throughout). Next, the ion motion is excited using two sets of SDKs separated by time $T$, with the first creating a cat state and the second reversing the process. After the second Ramsey microwave pulse of variable phase, the resulting interference is measured in the qubit population[6, 8] by applying a resonant laser to the ion and collecting qubit state-dependent fluorescence[25]. This sequence is detailed in the upper part of Fig. 2a.

The SDK originates from transform limited ultrafast laser pulses of duration $\tau \approx 10$ ps (or bandwidth $1/\tau \approx 100$ GHz) and centre optical wavelength $2\pi/k = 355$ nm. Each pulse enters an optical 50/50 beam splitter and is directed to arrive at the ion simultaneously ($\pm 70$ fs) in counter-propagating directions along $x$ and with orthogonal linear polarisations which are both orthogonal to an applied static magnetic field (Fig. 2a, lowest box). This produces a polarisation gradient at the ion and couples the qubit and ion motion with a sinusoidal modulation along the $x$-direction[26]. The bandwidth of each pulse is much larger than the hyperfine structure ($1/\tau \gg \omega_{\mathrm{hf}}$) but is narrow enough not to resonantly excite any higher energy states; the centre wavelength is detuned from the $^2P_{3/2}$ and $^2P_{1/2}$ levels (linewidth $\gamma/2\pi \sim 20$ MHz) by 67 and 33 THz, respectively. The resulting Raman process (Fig. 2b) gives rise to the single-pulse Hamiltonian[26]

$$\hat{H}(t) = \Omega(t)\sin\left[2kx_0\left(\hat{a}^\dagger + \hat{a}\right) + \phi\right]\hat{\sigma}_x + \frac{\omega_{\mathrm{hf}}}{2}\hat{\sigma}_z, \quad (1)$$

where $\hat{\sigma}_{x,z}$ are Pauli spin operators, $\phi$ is the relative phase between the counter-propagating light fields and is considered constant during a pulse, $\hat{a}^\dagger$ and $\hat{a}$ are the raising and lowering operators of the ion motion along $x$. We take the laser pulse shape as a hyperbolic secant function, with Rabi frequency $\Omega(t) = (\Theta/2\tau)\mathrm{sech}(\pi t/\tau)$ and pulse area $\Theta$, although the particular form of

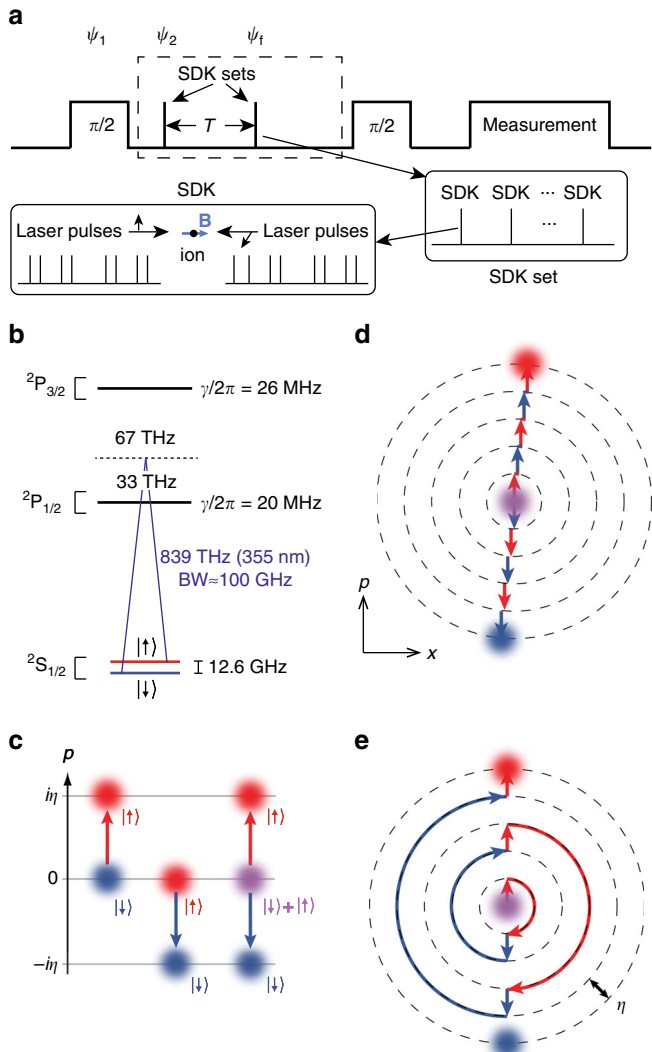

**Fig. 2** Experimental schedule and coherent state control. **a** Spin-dependent kicks (SDKs) are concatenated to generate cat states. The kicks are inserted between two Ramsey microwave $\pi/2$ pulses with a variable relative phase, and finally the interference of the motional states, after being separated for a time $T$, is measured on the qubit state. Counter-propagating pulses have linear orthogonal polarisations and are orthogonal to an applied magnetic field **B**. **b** Each SDK originates from position-dependent stimulated Raman transitions driven by counterpropagating laser beams near a wavelength of 355 nm. **c** A single SDK displaces a coherent state by $\pm 2\hbar k$ in momentum space (or $\pm i\eta$ in natural units of $\hbar/x_0$), with the sign depending on the initial qubit state (*red*: $|\uparrow\rangle$ or *blue*: $|\downarrow\rangle$), and splits a qubit superposition (*purple*: $|\uparrow\rangle + |\downarrow\rangle$). Each momentum transfer is accompanied by a spin flip. **d** SDKs are concatenated by changing the direction of the laser beams between each kick, with every pulse from a mode-locked laser used to increase the cat state separation. **e** Alternatively, SDKs are concatenated by timing the kicks to occur at each half-period of oscillation of the ion in the trap. (The *arrows* only track population initially in $|\downarrow\rangle$, but the final state is still shown by including the *blue* $|\downarrow\rangle$ component.) Free evolution appears in figures **d**, **e** as circular orbits

the pulse is not critical[26]. The single pulse interaction described in Eq. 1 is similar to Kapitza–Dirac scattering[22] through which the atomic motional wavepacket diffracts from a light field grating into all momenta classes $2n\hbar k$ with relative populations given by Bessel function $J_n(\Theta)$ of integer order $n$[26].

A single SDK is created by dividing each laser pulse above into a time sequence of eight sub-pulses using three stacked Mach-Zehnder interferometers[26]. Each of these eight sub-pulses has pulse area $\Theta \approx \pi/8$ with appropriately chosen phases $\phi$ determined by the ~100 ps (~3 cm) delays between the sub-pulse arrivals and a global frequency shift between the counter-propagating beams[26]. The net result is an SDK of momentum transfer of $\pm 2\hbar k$ for states $|\downarrow\rangle$ and $|\uparrow\rangle$, respectively[22, 24, 26], following the evolution operator

$$\hat{U}_{SDK} = \hat{\sigma}_+ \hat{\mathcal{D}}[i\eta] + \hat{\sigma}_- \hat{\mathcal{D}}[-i\eta]. \qquad (2)$$

In this expression, $\hat{\sigma}_\pm$ are the qubit raising and lowering operators, $\eta = 2kx_0 = 0.2$ is the Lamb-Dicke parameter associated with the momentum transfer, and $\hat{\mathcal{D}}$ is the phase-space displacement operator[18]. A remarkable feature of this interaction is that it does not rely on the LDR[26], where $\eta\sqrt{2n+1} \ll 1$, and therefore does not depend on tightly confined initial and final states.

Figure 2c depicts the SDK process in which the coherent state is shown in position–momentum phase space as a Gaussian disk and the colour represents the associated qubit state (the scale of the superposition states is drawn for illustrative purposes and are not scale). Each momentum displacement is associated with a qubit flip and has a fidelity of approximately 0.99. This SDK operation can be concatenated (Fig. 2d, e) to generate larger cat state separations, which remain in the harmonic potential region for $|\alpha| \lesssim d/x_0 \sim 10^4$, where $d \approx 100\,\mu m$ is the characteristic trap size.

**Large two-component cat states.** In the first of three experiments, we demonstrate a fast method for generating large cat states by concatenating $N$ SDKs with successive laser pulses from a 355 nm mode locked laser (repetition rate $f_{rep} = 81.4$ MHz). This is achieved by alternating the directions (by swapping the paths of propagation) of the counter-propagating beams for each successive pulse using a Pockels cell (see Methods and Fig. 2d). In this way, the cat state separation grows with an average rate $\frac{d(\Delta\alpha)}{dt} \approx 2\eta f_{rep}$, ideally generating the cat state

$$|\psi_2\rangle = |\uparrow\rangle|\alpha\rangle + |\downarrow\rangle|-\alpha\rangle, \qquad (3)$$

where $\alpha = iN\eta$. After allowing the state to evolve for varying amounts of time $T$, the reversal step ideally generates the state $|\Psi_f\rangle = |\uparrow\rangle|-\alpha e^{-i\theta} + \alpha\rangle + |\downarrow\rangle|\alpha e^{-i\theta} - \alpha\rangle$, where $\theta = \omega T$ (Fig. 3a). When the phase of the second Ramsey microwave $\pi/2$ pulse is scanned, the resulting interference contrast in the qubit population can be written as[26]

$$C(\theta) = C_0 e^{-4|\alpha|^2(1-\cos\theta)}, \qquad (4)$$

where $C_0 < 1$ accounts for imperfect operations. When the delay $T$ is near an integer multiple $m$ of the trap period ($\theta \sim 2\pi m$), we observe a revival in the Ramsey contrast, and for $|\alpha| \gg \frac{1}{\sqrt{2}}$, the shape of the contrast revival is approximately Gaussian with an expected FWHM of $\Delta\theta \approx 1.18/|\alpha|$[24]. In Fig. 3b, revival lineshapes at $\theta = 2\pi$ are shown in which the state $|\psi_2\rangle$ is generated for various times up to $\Delta\alpha = 4.0$ in 111 ns (upper plot). The data fit well to the functional form of Eq. 4 with the peak contrast $C_0$ as the only fit parameter. The cat state fidelity, estimated using the relation $F = C_0^{1/2}$, decays from ~90 to 60% as the cat state is made bigger (lower plot). These data are consistent with an effective single SDK fidelity of 0.951(4), which is lower than that of an isolated single SDK because of power fluctuations associated with swapping of successive laser pulses. Despite the lower fidelities of this technique, it is an important benchmark for ultrafast quantum gates with trapped ions[27, 28].

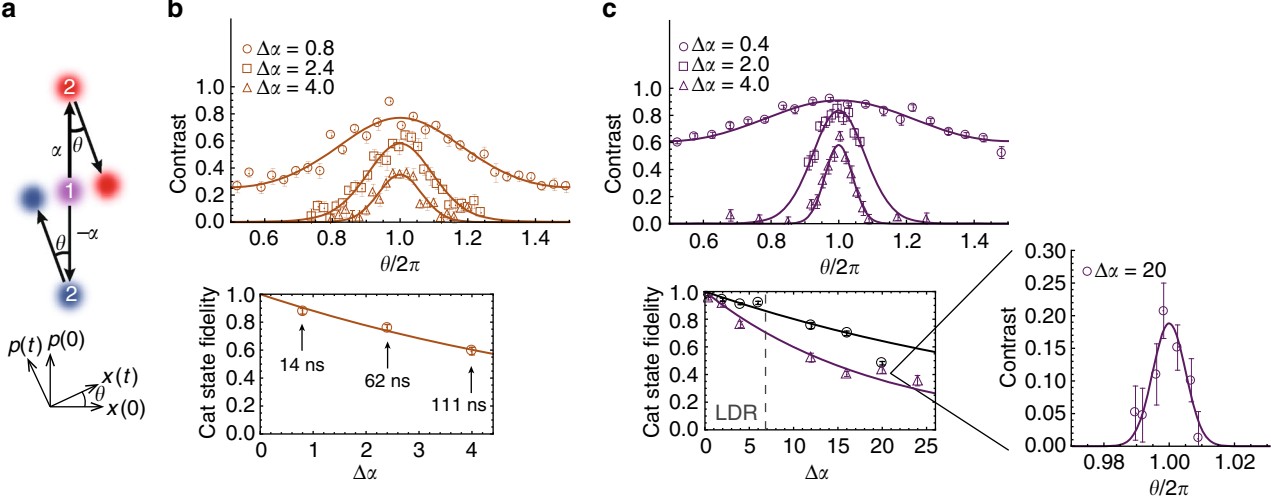

**Fig. 3** Cat state creation and verification. **a** The state $|\psi_1\rangle = |\uparrow\rangle + |\downarrow\rangle$ (labelled "1") is split using a set of SDKs to create the cat state $|\psi_2\rangle = |\uparrow\rangle|\alpha\rangle + |\downarrow\rangle|-\alpha\rangle$ ("2") that has a separation of $\Delta\alpha$ between its components. After evolution $\theta = \omega T$ ($\omega/2\pi = 1.0$ MHz), a second set of SDKs drives the state to $|\Psi_f\rangle = |\uparrow\rangle|-\alpha e^{-i\theta} + \alpha\rangle + |\downarrow\rangle|\alpha e^{-i\theta} - \alpha\rangle$. **b** Switching each successive laser pulse as an SDK, the cat state $|\psi_2\rangle$ with $\Delta\alpha = 0.8$ is generated in about 14 ns, $\Delta\alpha = 2.4$ in 62 ns and $\Delta\alpha = 4.0$ in 111 ns. The states are verified by observing contrast in the state $|\Psi_f\rangle$ (upper plot). We find the fidelity of each cat state $|\psi_2\rangle$ to be 0.88(2), 0.76(2) and 0.59(3), respectively (lower plot). **c** Using the evolution of the atom in the trap to swap SDKs, the generation is slower but has higher fidelity because the laser beam paths are not alternating. The effective single SDK fidelities are 0.9912 and 0.98 for Doppler (black circles) and ground state (purple triangles) cooled atoms (lower plot; the dashed line in the lower plot signifies the limit of the Lamb-Dicke Regime (LDR)). These states are generated in times of $\sim (\Delta\alpha - 0.4) \times 1250$ ns. Again, the states are verified by observing contrast in the state $|\Psi_f\rangle$ (upper plot). The inset shows a cat state with separation $\Delta\alpha = 20$ and revival peak contrast of $C_0 = 0.19(3)$. In **b**, **c**, error bars are statistical with confidence interval of $\pm$ one standard deviation. The solid lines are fits to the underlying theory (Eq. 4, with the peak contrast as the only fit parameter)

In a second set of experiments, we create larger cat states by delivering an SDK at every half trap period instead of switching laser beam paths (Fig. 2e). This maintains a high SDK fidelity by leaving the beam paths stationary, while the cat state grows at an average rate of $\frac{d|\alpha|}{dt} = \eta\omega/\pi$. By reversing the cat generation as above, we produce and verify states up to $\Delta\alpha = 24$. Again, the data fit well to the functional form of Eq. 4 with the peak contrast as the only fit parameter (Fig. 3c). Contrast of $C_0 = 0.19$ (2) is measured for the state with $\Delta\alpha = 20$ (inset of Fig. 3c), which has a maximum momentum separation of $200\hbar k$ between coherent states and maximum spacial separation of 209 nm. As a comparison to an unconfined system, ref. [7] generates $90\hbar k$ of momentum separation between components of a cold atomic gas and achieves a spatial separation of 54 cm after allowing the components to drift apart for 1 s. In our apparatus, such large superposition states gives rise to very narrow interference patterns and requires a high level of trap stability, which is achieved using a RF stabilisation procedure[23]. For these largest cat states, we scan the trap frequency $\omega$ for fine control in the trap evolution phase $\theta$[24]. From these data, we again infer the fidelity of each single SDK, which is 0.980(1) for displacing states initially cooled to near the ground state, and 0.991(1) for states initially cooled to the Doppler limit[24]. The lower fidelity stems from the slower data collection rate due to the dwell time of ground state cooling, increasing susceptibility to drifts in the trap frequency (see Methods).

**Multicomponent cat states**. The speed, fidelity and high level of control in ultrafast displacement operations allows us to prepare more complicated, multicomponent states. First, we create three- and four-component states with one additional microwave pulse and SDK set. Starting from the state $|\psi_2\rangle$, a microwave $\pi/2$ pulse rotates the state to $|\psi_3\rangle = (|\uparrow\rangle - |\downarrow\rangle)|\alpha\rangle + (|\uparrow\rangle + |\downarrow\rangle)|-\alpha\rangle$. A set of SDKs then produces three- and four-component superposition

states of the form

$$
\begin{aligned}
|\Psi_{cat}^{3,4}\rangle = &|\uparrow\rangle\left(e^{i\phi_1}|\alpha e^{-i\theta} + \alpha\rangle + e^{i\phi_2}|\alpha e^{-i\theta} - \alpha\rangle\right) \\
&+ |\downarrow\rangle\left(e^{i\phi_3}|-\alpha e^{-i\theta} + \alpha\rangle + e^{i\phi_4}|-\alpha e^{-i\theta} - \alpha\rangle\right),
\end{aligned} \quad (5)
$$

with configuration depending on the phase delay $\theta$ (Fig. 4a). (phases $\phi_1$, $\phi_2$, $\phi_3$ and $\phi_4$ are defined in Methods). It is evident from Eq. 5 that a three-component state is created when $\theta = m\pi$, and a four-component state is generated for other values of $\theta$. Scanning $\theta$ and the phase of a final Ramsey microwave $\pi/2$ pulse, we observe a contrast lineshape indicative of the desired state (Fig. 4b). To further verify that these multicomponent states are being created, we run the same sequence but apply either no microwave pulse, or a $\pi$ pulse, to the state $|\psi_2\rangle$. An SDK set then generates the states $|\Psi_{cat,0}\rangle = |\uparrow\rangle|-\alpha e^{-i\theta} + \alpha\rangle + |\downarrow\rangle|\alpha e^{-i\theta} - \alpha\rangle$ and $|\Psi_{cat,\pi}\rangle = |\downarrow\rangle|\alpha e^{-i\theta} + \alpha\rangle + |\uparrow\rangle|-\alpha e^{-i\theta} - \alpha\rangle$. These states revive at the same phase delay $\theta$, but out of phase by $\pi$, which is verified in Fig. 4c.

Continuing to unfold the state in phase space, another microwave $\pi/2$ rotation and SDK set generates a six- and eight-component state (Fig. 4d). In this case, the four-component state is generated with a separation along one quadrature that is twice as large as the other, allowing for a square lattice once the eight-component state is created. Again, scanning the phase delay $\theta$ and the phase of a final microwave pulse, Ramsey fringes are observed which compare well with the expected theoretical behaviour (Fig. 4e). Because of the complexity of the final state, equations for this superposition state are left to the Methods section. Each SDK set contains two SDKs in these multi-component state experiments, meaning when the components are in a square lattice, the nearest-neighbour component separation is $\Delta\alpha \approx 0.8$. These states are close enough to overlap as they evolve in phase space, allowing us to retrieve the characteristic signals plotted in Fig. 4. Multicomponent states with larger phase–space

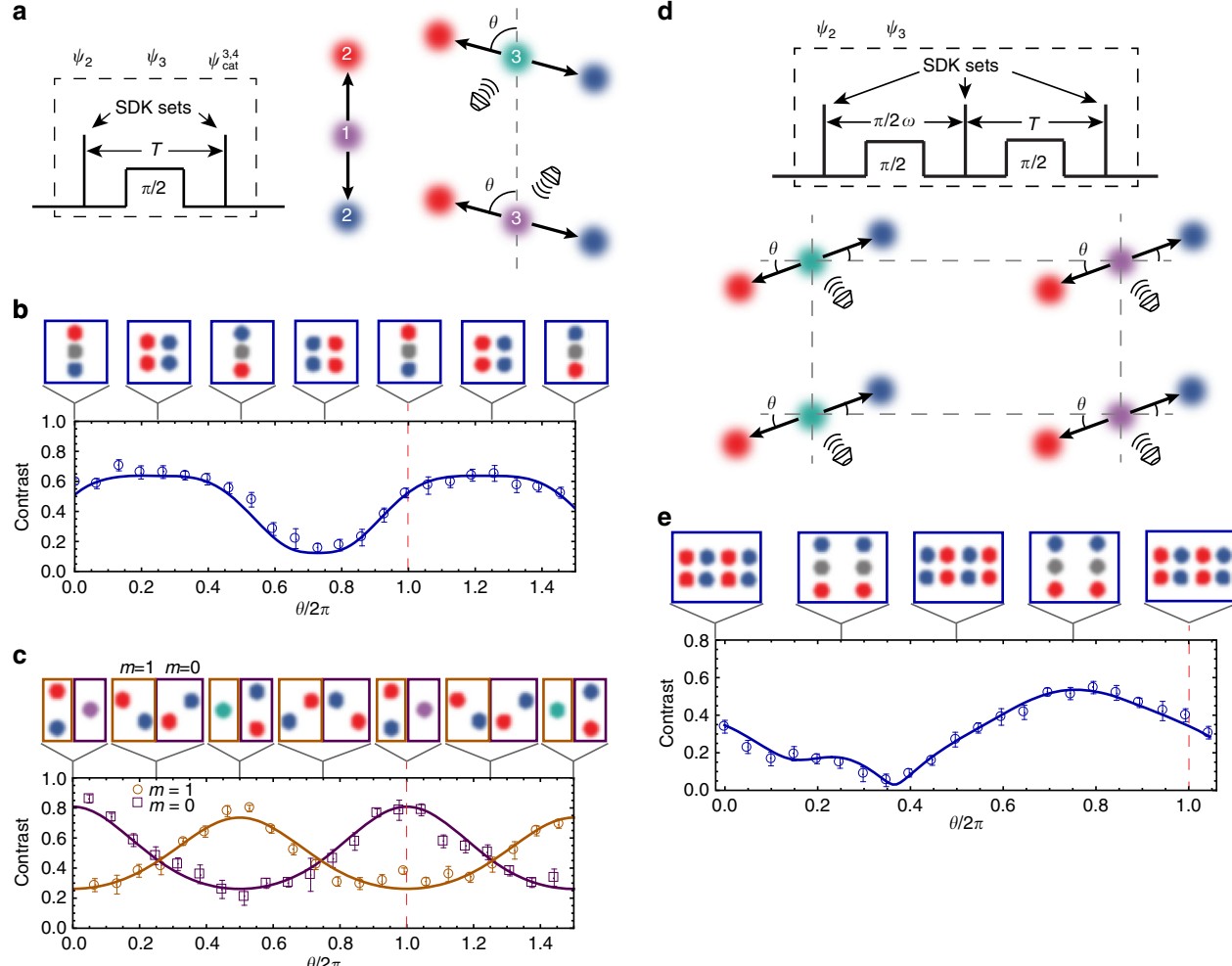

**Fig. 4** Three-, four-, six- and eight-component states. **a** Creation of a multicomponent state begins by applying a set of SDKs to take the state $|\psi_1\rangle$ (1) to the state $|\psi_2\rangle$ (2). A microwave $\pi/2$ pulse rotates the qubit to produce the state $|\psi_3\rangle = (|\uparrow\rangle - |\downarrow\rangle)|\alpha\rangle + (|\uparrow\rangle + |\downarrow\rangle)|-\alpha\rangle$ (3). Another set of SDKs generates the three- or four-component state. The diagram within the *dashed box* replaces the one in Fig. 1a for these experiments. **b** If $\theta = 0$, two of the components rejoin and the state has the form $|\alpha\rangle + |0\rangle + |-\alpha\rangle$. If $\theta = \pi/4$, for instance, then a four-component state of the form $|\alpha\rangle + |-\alpha\rangle + |i\alpha\rangle + |-i\alpha\rangle$ is generated. These configurations are depicted in the flags above the contrast curve. The final microwave pulse analyses the state contrast, and is plotted as a function of $\theta$, which is compared with the predicted contrast curve with only the amplitude as a fitting parameter. *Error bars* are calculated with confidence interval of one sigma. **c** If the microwave $\pi/2$ pulse in **a** is replaced by a $m\pi$ pulse, then the second SDK set behaves as it would in the two-component experiment, with the exception that odd values of $m$ are shifted by half of a trap period. We see this behaviour fits the predicted model well in the figure with $m = 0$ (*purple*), and $m = 1$ (*gold*). **d** The six- and eight-component state is created by extending the technique for the three- and four-component state with an additional microwave pulse and SDK set. **e** Contrast as a function of $\theta$ is used to verify the creation of the superposition state when compared to the model (*solid line*)

separations can be made in the same way; however, measuring the details of the state may require more complete tomographic techniques.

## Discussion

Ultrafast laser pulses are capable of generating Schrödinger cat states much larger than presented here, theoretically limited not by the Lamb-Dicke limit, but by the size of the laser beam. In addition, if the superposition state is extended by relying on free evolution in the trap, dispersion from anharmonic motion will also limit the size of the superposition. This technique can also be used to make more complicated multicomponent states, as well as generate them in two and three dimensions by modifying the trapping potential and orientation. In order to generate larger separations, the trap frequency could be substantially lowered: operating the current apparatus at $\omega/2\pi = 10$ kHz could produce cat state separations as large as 20 µm. This would enhance the

sensitivity interferometric measurements of rotation[29] or proximal electric field gradients. At such large separations, we can directly resolve the cat components with high-resolution imaging techniques[30], allowing investigations in measurement backaction and Heisenbergs microscope type thought experiments[31].

## Methods

**Experimental setup**. Laser pulses are generated from a frequency tripled, mode-locked Nd:YVO$_4$ laser. The laser repetition rate $f_{rep} = 81.4$ MHz is not actively stabilised, and exhibits a drift of about 10 Hz over 1 min (or a 0.8 µrad min$^{-1}$ drift in $\theta$ at one trap period, which is insignificant for all data presented here). The spin-dependent displacement in Eq. 2 has the full form:

$$\hat{O}_{SDK} = e^{i\phi_\lambda}\hat{\sigma}_+\hat{\mathcal{D}}[i\eta] + e^{-i\phi_\lambda}\hat{\sigma}_-\hat{\mathcal{D}}[-i\eta], \quad (6)$$

where the phase $\phi_\lambda$ is an optical phase that is stable during the course of one experiment, but random over multiple experiments due to repetition rate drift and slow mechanical and other drifts in the optical path. However, the effect of the

phase $\phi_\lambda$ cancels when an even number of applications of the operator $\hat{O}_{SDK}$ are used during an experiment and so the optical phase terms are dropped in Eq. 2.

The first method discussed for generating cat states uses every pulse from the mode-locked laser (Fig. 2d). This works by swapping the directions of the counter-propagating beams, countering the spin flip that occurs with each SDK. To make this swap, we combine the perpendicular linearly polarised beams on a polarising beam splitter and pass them through a Pockels cell. The cell can rotate the polarisations by 0 or $\pi/2$ radians arbitrarily for pulses arriving every 12 ns; here we alternate every pulse. A polarising beam cube downstream of the Pockels cell separates the two beams after which they are directed, counter-propagating, onto the ion with simultaneous arrivals. The rate at which the cat state grows, $\frac{d(\Delta\alpha)}{dt} \approx 2\eta f_{rep}$, holds only for the number of kicks $N \ll 2\pi f_{rep}/\omega$, which is the case in the experiment. For larger numbers of kicks, the growth rate is expected to decrease as the trap evolution reverses the kick direction. In the future this could be compensated by adding an extra beam reversal each half trap period.

**Three and four-component cat contrast.** The contrast function that overlays the data in Fig. 4b is derived here. We write the time evolution operator for a coherent state as $\hat{U}_T[\theta]|\alpha\rangle = |\alpha e^{-i\theta}\rangle$. The microwave rotation operator in the $z$-basis is written as

$$\hat{R}_\mu[\phi_\mu] = \frac{1}{\sqrt{2}}\hat{\mathbb{1}} \otimes \begin{bmatrix} 1 & e^{i\phi_\mu} \\ -e^{-i\phi_\mu} & 1 \end{bmatrix}, \qquad (7)$$

where all rotations have pulse area $\pi/2$. A full Ramsey experiment to create three- and four-component cat states, including microwave rotations, SDKs, free evolution and a final analysis microwave pulse produces the final state

$$\begin{aligned} \left|\Psi_f^\beta\right\rangle = &\hat{R}_\mu\left[\phi_\mu'''\right] \cdot \hat{O}_{SDK} \cdot \hat{U}_T[\pi] \cdot \hat{O}_{SDK} \cdot \hat{U}_T[\theta] \cdot \\ &\hat{R}_\mu\left[\phi_\mu''\right] \cdot \hat{O}_{SDK} \cdot \hat{U}_T[\pi] \cdot \hat{O}_{SDK} \cdot \hat{R}_\mu\left[\phi_\mu'\right] \cdot |\downarrow\rangle|\beta\rangle. \end{aligned} \qquad (8)$$

The spin-up portion of the final state is given as

$$\begin{aligned} &\exp\left(-2i\eta\beta_R + 2i\eta\,\text{Re}\left[e^{-i\theta}(2i\eta - \beta)\right] + i\phi_\mu'' - i\phi_\mu' - i\phi_\mu'''\right)\left|-2i\eta - e^{-i\theta}(2i\eta - \beta)\right\rangle \\ &-\exp\left(-2i\eta\beta_R - 2i\eta\,\text{Re}\left[e^{-i\theta}(2i\eta - \beta)\right] - i\phi_\mu'\right)\left|2i\eta - e^{-i\theta}(2i\eta - \beta)\right\rangle \\ &-\exp\left(2i\eta\beta_R - 2i\eta\,\text{Re}\left[e^{-i\theta}(-2i\eta - \beta)\right] - i\phi_\mu''\right)\left|2i\eta - e^{-i\theta}(-2i\eta - \beta)\right\rangle \\ &-\exp\left(2i\eta\beta_R + 2i\eta\,\text{Re}\left[e^{-i\theta}(-2i\eta - \beta)\right] - i\phi_\mu'''\right)\left|-2i\eta - e^{-i\theta}(-2i\eta - \beta)\right\rangle, \end{aligned} \qquad (9)$$

where the normalisation factor and spin-up ket is left out for simplicity. The brightness for any thermal state with average phonon occupation $\bar{n}$ is given as

$$B = \frac{1}{\pi\bar{n}}\int_{-\infty}^{\infty} e^{-|\beta|^2/\bar{n}}\left\langle\uparrow\left|\Psi_f^\beta\right\rangle\left\langle\Psi_f^\beta\right|\uparrow\right\rangle d^2\beta. \qquad (10)$$

For an ion initially in a thermal motional state the brightness is

$$\begin{aligned} &\tfrac{1}{4}\left[1 + e^{16(1+2\bar{n})\eta^2(\cos\theta-1)}\cos\left(\phi_\mu' - \phi_\mu'''\right)\right] \\ &+\tfrac{1}{4}\left[1 - e^{-32(1+2\bar{n})\eta^2\cos^2\left(\frac{\theta}{2}\right)}\cos\left(2\phi_\mu'' - \phi_\mu' - \phi_\mu'''\right)\right] \\ &+\tfrac{1}{\sqrt{8}}e^{-8(1+2\bar{n})\eta^2}\sin\left(16\eta^2\sin\theta\right)\sin\left(\phi'' - \phi_\mu'''\right). \end{aligned} \qquad (11)$$

**Six- and eight-component cat contrast.** This calculation is carried out in the same fashion, using the full set of operations

$$\begin{aligned} \left|\Psi_f^\beta\right\rangle = \ &\hat{R}_\mu\left[\phi_\mu''''\right] \cdot \hat{O}_{SDK} \cdot \hat{U}_T[\pi] \cdot \hat{O}_{SDK} \cdot \hat{U}_T[\theta] \\ &\cdot\hat{R}_\mu\left[\phi_\mu'''\right] \cdot \hat{O}_{SDK} \cdot \hat{U}_T[\pi] \cdot \hat{O}_{SDK} \cdot \hat{U}_T[\pi] \\ &\cdot\hat{O}_{SDK} \cdot \hat{U}_T[\pi] \cdot \hat{O}_{SDK} \cdot \hat{U}_T\left[\tfrac{\pi}{2}\right] \cdot \hat{R}_\mu\left[\phi_\mu''\right] \\ &\cdot\hat{O}_{SDK} \cdot \hat{U}_T[\pi] \cdot \hat{O}_{SDK} \cdot \hat{R}_\mu\left[\phi_\mu'\right] \cdot |\uparrow\rangle|\beta\rangle. \end{aligned} \qquad (12)$$

We do not show the full brightness calculation here because of its length. The solid line in Fig. 4e is a fit assuming that the initial motional state is $\beta = 0$. Our initial thermal occupation number is $\bar{n} = 0.15$, or about 87% in the ground state. We do not take the thermal average of this expression, owing to computational complexity. As with the lineshape for the three- and four-component cat state (Fig. 4b, c), we used the contrast peak amplitude as the only fitting parameter.

**Sources of error.** Several factors lead to imperfect fidelity of the cat states we create. As the cat states are made larger, their interference fringes are narrower, with increased susceptibility to a host of drifts.

The trap axes are rotated so that the Raman beam nominally couples only to a single mode. We estimate a misalignment of <0.5°, which entangles <0.8% of the qubit population with other perpendicular modes of motion for each SDK. The alignment can be made to have a tighter tolerance, likely by at least a factor of 10 by using more sophisticated alignment methods. Detection fidelity of the qubit is 99%. We do not expect that trap anharmonicities contribute to a loss in interference contrast.

The Raman beam waist is ≈3 µm, with a divergence of about 2°; the Rayleigh distance is 80 µm, meaning that the ion would need to venture a considerable distance farther than it does during its excited oscillations in order to see an appreciable fraction of the divergence change in kick direction. Additionally, the axial component of a beam at its focus (with a waist of 3 µm) is less than about 0.001% and is negligible.

While the initial motional state does not affect the fidelity of an SDK, ground state cooling increases data acquisition time by a factor of about seven times. This increases the acquisition time of a single, 10 point contrast lineshape from about 20 to 140 s. While it is difficult to accurately assess the behaviour of trap frequency noise on that time scale, trap frequency drifts on the order of kilohertz would cause the difference in fidelity seen between the revival lineshape of large cat states made from ground state cooled oscillations and Doppler cooled only states. Based on standard noise models, the 7× increase in acquisition time is likely the cause of this difference.

**Data availability**. The data sets generated and analysed during this study are available from the corresponding author on reasonable request.

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

## Acknowledgements

We thank S. Moses for comments on the manuscript. This work is supported by the Army Research Office and NSF Physics Frontier Center at JQI.

## Author contributions

All authors contributed to the design, construction and carrying out of the experiment, discussed the results and commented on the manuscript. K.G.J. and J.D.W.-C. analysed the data and performed the simulations. K.G.J., J.D.W.-C. and C.M. wrote the manuscript.

## Additional information

**Competing interests:** The authors declare no competing financial interests.

