## [Peer Review File · Nature Communications]

Reviewers' comments:

Reviewer #1 (Remarks to the Author):

Dear editor,

The manuscript by Johnson and co-workers describes several experiments in which a series of ultrafast, off-resonance Raman transitions were used to generate very large ($D \sim 24$) Schrödinger cat states of the spin and motion of a single trapped ion. While cat states of single ions, and other harmonic oscillators, were demonstrated before, these experiments are impressive as they claim to demonstrate the largest Cat so far. Furthermore, this is the first time a multi-component cat state is demonstrated using a mechanical oscillator. Last, and in my opinion most importantly, this cat state is generated using ultrafast pulses and is an important benchmark on the way to demonstrating high fidelity ultrafast entanglement gates in trapped ion systems. This paper is therefore of very high quality and certainly deserves to be published in Nature communications.

I only have very few minor comments:

1. In page 6 line 11 the authors say that when $|\alpha| \ll 1$ unharmonicities do not play a role. I am not sure I understand how unharmonicities would interfere with this method. Usually I think of small un-harmonicity as a small shift in the resonance frequency between adjacent h.o. levels. Here clearly this would not matter as the pulse time is way too short to differentiate between such levels. A short sentence on how oscillator non-linearity would affect this method would be helpful.
2. A small thing but confused me: in Fig. 2 the authors sometime use $|\alpha|$ and sometimes (along the x axis) $\Delta\alpha$. I would conform to one or the other to make the comparisons easier.
3. In a recent experiment the interference pattern between two cat components with $\Delta\alpha \sim 15$ was measured by the ETH group on a single trapped ion. I think this paper (PRL 116, 140402 (2016)) is highly relevant and should be cited.

Reviewer #2 (Remarks to the Author):

The authors demonstrate the creation of Schrödinger cat states in a tightly confined atom through mapping a hyperfine spin superposition into a spatial super-position through the use of precisely timed state dependent kicks (SDKs). That is to say they've observed the interferometric signature of this effect when written back into the qubit state. The experimental work is well done, however there are issues with the context in which this work is placed and portions of the results and the experimental technique could be communicated with better clarity to non ion trapping specialists. A definite weakness of the manuscript is the indirect readout of the more complex spatial superpositions, particularly in light of Ref 30, where the same group (though perhaps a different experimental apparatus?) achieved nanometer scale localisation of Yb^+ ions. With a maximum spatial separation of 209 nm, I would have hoped some rough spatial information might be recoverable by deconvolution techniques.

Specific concerns

With regards to potential applications in sensing, spatial superpositions are the basis for atom interferometry, the developments which have occurred since 1991 (Ref 6) are substantial, certainly exceed the level of mere "potential", and should be acknowledged in a bit more depth. In particular this manuscript would benefit from referencing

T. Kovachy, P. Asenbaum, C. Overstreet, C. A. Donnelly, S. M. Dickerson, A. Sugarbaker, J. M. Hogan and M. A. Kasevich, Quantum superposition at the half-metre scale, Nature, 528, 530–533 (2015).

Comparison to this paper is apt as it used $90\hbar k$ of momentum, while the authors have $100\hbar k$ of momentum in each coherent state. This brings out the difference between a tightly confining trap environment where a maximum spatial separation of 209 nm was achieved vs. a freely expanding cold neutral atomic gas with its much larger length scale. The free space experimental systems are different from the authors and the trapped environment offers potential advantages in terms of persistence.

Line 30-, "the natural localised state is a coherent state." This could be phrased more understandably in terms of expressing that a coherent state is a displaced (in position, momentum, or some combination of both) quantum ground state. This makes more sense as a route to justifying it as natural localised quantum state and makes discussions of coherent superposition states more intuitive with regards to spatial or momentum displacements. It would also do well here to introduce the spatial/momentum tradeoff that the trapping provides. As opposed to the one way street that occurs in free expansion/time of flight type experiments in neutral atoms.

Line 36- Could the authors please clarify why they've chosen to cite Ref 18 (Wineland Nobel Prize Review Mod Phys paper) regarding creating cat states in a massive particle instead of the original work by the last author C. Monroe*, D. M. Meekhof, B. E. King, D. J. Wineland A "Schrödinger Cat" Superposition State of an Atom Science 1996 Vol 272 pp 1131-1136. There is also the issue of the use of the singular particle vs. plural particles in regards to this statement as the work in question was using a single 9Be^+ ion. Using another/additional citation where multiple massive particles were in a cat state might also be appropriate here.

Lines 43-45 - Could be more clear, particularly given that Ref 21's abstract explicitly states that they "experimentally observe the behaviour outside the Lamb-Dicke regime."

Line 79- What is the wavelength spread of this laser pulse? Does it hit the $P_{3/2}$ state at all? What do you mean does not couple directly to any of the excited states? What is the excitation probability for driving a spontaneous decay? Its the wavelength spread, rather than the timing, which determines if the $P_{1/2}$ and $P_{3/2}$ state are excited. Alternatively stating something about the time-bandwidth product of the pulse would also indicate how Fourier transform limited its behaviour is.

Line 82- "arrive at the ion simultaneously" With regards to a 10 ps pulse. Are they within 0.1 ps of each other? 1 ps of each other? What is the tolerance here. The word "simultaneously" makes me nervous as a physicist when dealing with laser pulses. It would also be helpful to observe that the pulses are $\sim 1/1000$ the excited state lifetime of the P state lifetime, for those not familiar with the specifics of Yb^+ 's level structure.

Line 86- Clarify what sort of pulse the polarisation is creating. I understand its lin perp lin, but what is the B field orientation in the system. E.g. are these all driving Π transitions, Σ^+ , Σ^- , etc.? Please add a brief clarification for those not familiar with the details in Ref 26 regards how the state dependent force comes about (pulse ordering of absorb/emit with respect to state).

Figure 1b- the red vs. blue in the ground hyperfine splitting is barely visible. Thicken this up!

Figure 1c- Are you intending to show the spin dependent kick also flips the spin state?

Figure 1d. Where are the arrows in the lower half of the diagram to show the spin dependent kicks increasing the "orbital radius"

Figure 1d,e. Would be good to remind people here that $\text{Im}(\alpha)$ is also momentum and

Re(α) is position.

Figure 1 caption. Typo.

"the kicks to occur at each half-period of oscillation of the ion in trha trap."

"the kicks to occur at each half-period of oscillation of the ion in the trap."

Line 103-104- "A remarkable feature of this interaction is that it does not rely on confinement to the Lamb-Dicke regime"- Does this mean that it wasn't necessary to cool to near the ground state in order to perform this experiment? Or does it just change the fidelity

Line 111, For context What are the dimensions of your trap in the direction you're kicking.

Line 115- by directions do you mean polarisation directions?

Line 154 "Multicomponent superposition states have not previously been created in the motion of atoms."- This statement is not correct in the context of atom-interferometry and trapped neutral atom BEC type experiments. See "Optics and interferometry with atoms and molecules" Alexander D. Cronin, Jörg Schmiedmayer, and David E. Pritchard Rev. Mod. Phys. 81, 1051 (2009) For instance in Fig 2 dating back to 1930 shows atom diffraction peaks, thus a superposition of three states in the motion of atoms.

Figure 2 caption- Mention here that the period $T=1000$ ns to given context for the 62 ns to get $\alpha=1.2$ and 111 ns to get $\alpha=2.0$.

Line 172- Please clarify what you mean by larger? Larger momentum difference, or larger spatially. Though these are linked, larger spatially also has to do with confinement and trap anharmonicity. E.g. is it larger α that is of interest, or larger enclosed area.

Line 179- The authors might do well to speculate as to the applications of such large cat states to a more extended range of problems in quantum gravity or a Heisenberg's microscope type problems.

184-185- What are the consequences of this laser's rep rate not being actively stabilised. I'll note its frequency drift is on the order of what I'm guessing is the the actively stabilised trap frequency drift in Ref 23. Again this depends on if its the same or a similar experimental apparatus.

222- What is the divergence of the focused Raman beams? Will this spread in angles also couple to the other directions? I would be concerned that this spread in angle is more important than the misalignment.

225- Do you have an estimate of the trap anharmonic length scale(s). E.g. the ratio of the 2nd order quadratic confinement driving potential curvature term vs. the 3rd order cubic potential coefficient or 4th order quartic term. I'm assuming your electrodes are not precisely crafted conic section.

Minor corrections

Capitalisation of paper titles

Ref 6 - Raman

Ref 7,13,18 - Schrödinger

Ref 21- Lamb-Dicke

Typo

Ref 25 +Yb should be Yb+

Reviewer #3 (Remarks to the Author):

In their paper "Ultrafast create of large Schrodinger cats states of an atom" the authors present experiments generating and characterizing a variety of large spin-motion entangled states using a pulsed laser.

I think that this paper presents novel, interesting, results, and is well written. Once the authors address the below minor comments and questions I believe this work merits publication in Nature Communications.

* The phrase 'ultrafast' is used throughout the paper, yet the operations that create the large cat states are not that fast (tens of micro-seconds), due to repeated delays waiting for motional precession. I think the authors should explicitly mention the time-scales for creating the large entangled states using the trap evolution method, as they do for faster method in the caption of figure 2.

* Around line 148 the infidelity of the SDKs from trap frequency drift, with and without sub-doppler cooling, is discussed. It would be helpful for there to be some mention of the magnitude of frequency drift that would cause such an error, as well as the relative data collection time with and without sub-doppler cooling.

* In Methods (d) the angle between the Raman delta-k and the desired mode is said to be < 0.5 deg, and it is said that this will set an upper limit on the 8 component cat contrast of 58(3)% - this seems to be quite a small uncertainty. If this is not false precision, the measured mode projections used to calculate this should be stated.

* It seems like it might be relevant to cite PRL 116 140402 (2016)

Typographic:

* line 60: I assume this should be " $w = w_x, w_y, w_z$ " (not " $w_x = w, w_y, w_z$ ")

* caption of fig. 1, (e): trha -> the

* line 220: suscpetibility -> susceptibility

Reviewer #1 had the following to say:

Dear editor,

The manuscript by Johnson and co-workers describes several experiments in which a series of ultrafast, off-resonance Raman transitions were used to generate very large ($D \sim 24$) Schrodinger cat states of the spin and motion of a single trapped ion. While cat states of single ions, and other harmonic oscillators, were demonstrated before, these experiments are impressive as they claim to demonstrate the largest Cat so far. Furthermore, this is the first time a multi-component cat state is demonstrated using a mechanical oscillator. Last, and in my opinion most importantly, this cat state is generated using ultrafast pulses and is an important benchmark on the way to demonstrating high fidelity ultrafast entanglement gates in trapped ion systems. This paper is therefore of very high quality and certainly deserves to be published in Nature communications.

I only have very few minor comments:

With 3 comments:

1) “In page 6 line 11 the authors say that when $|\alpha| \ll 1$ unharmonicities do not play a role. I am not sure I understand how unharmonicities would interfere with this method. Usually I think of small un-harmonicity as a small shift in the resonance frequency between adjacent h.o. levels. Here clearly this would not matter as the pulse time is way too short to differentiate between such levels. A short sentence on how oscillator non-linearity would affect this method would be helpful.”

Response: The referee is correct; we are not limited by anharmonicities during the kick itself. However, the dispersion of motion during the free evolution is the problem. We have changed the phrase (now in line 119)

“...so long as the atomic motion remains confined within the harmonic trapping region, or...”

to

“...which remain in the harmonic potential region for...” to clarify that only the trajectory is affected, not the function of the kick.

We also modified line 186 from

“Ultrafast laser pulses are capable of generating Schrodinger cat states much larger than presented here, theoretically limited not by the Lamb-Dicke limit, but by the size of the laser beam and the anharmonicity of the trap.”

to

“Ultrafast laser pulses are capable of generating Schrodinger cat states much larger than presented here, theoretically limited not by the Lamb-Dicke limit, but by the size of the laser beam. In addition, if the superposition state is extended by relying on free evolution in the trap, dispersion from anharmonic motion will also limit the size of the superposition.”

2) “A small thing but confused me: in Fig. 2 the authors sometime use $|\alpha|$ and sometimes (along the x axis) $\Delta\alpha$. I would conform to one or the other to make the comparisons easier.”

Response: We have changed the separation parameter to be $\Delta\alpha$ in the plots in Fig. 2 (now Fig. 3). We have also adjusted phrasing in the legend to reflect this.

3) “In a recent experiment the interference pattern between two cat components with $\Delta\alpha \sim 15$ was measured by the ETH group on a single trapped ion. I think this paper (PRL 116, 140402 (2016)) is highly relevant and should be cited.”

Response: We have added a reference to PRL 116, 140402 (2016).

Reviewer #2 had the following to say:

The authors demonstrate the creation of Schrödinger cat states in a tightly confined atom through mapping a hyperfine spin superposition into a spatial super-position through the use of precisely timed state dependent kicks (SDKs). That is to say they've observed the interferometric signature of this effect when written back into the qubit state. The experimental work is well done, however there are issues with the context in which this work is placed and portions of the results and the experimental technique could be communicated with better clarity to non ion trapping specialists. A definite weakness of the manuscript is the indirect readout of the more complex spatial superpositions, particularly in light of Ref 30, where the same group (though perhaps a different experimental apparatus?) achieved nanometer scale localisation of Yb⁺ ions. With a maximum spatial separation of 209 nm, I would have hoped some rough spatial information might be recoverable by deconvolution techniques. *(This comment is addressed in the following section.)*

Specific concerns

With 23 comments:

1) “With regards to potential applications in sensing, spatial superpositions are the basis for atom interferometry, the developments which have occurred since 1991 (Ref 6) are substantial, certainly exceed the level of mere “potential”, and should be acknowledged in a bit more depth. In particular this manuscript would benefit from referencing T. Kovachy, et al., Nature, 528, 530–533 (2015). Comparison to this paper is apt as it used $90\hbar k$ of momentum, while the authors have $100\hbar k$ of momentum in each coherent state. This brings out the difference between a tightly confining trap environment where a maximum spatial separation of 209 nm was achieved vs. a freely expanding cold neutral atomic gas with its much larger length scale. The free space experimental systems are different from the authors and the trapped environment offers potential advantages in terms of persistence.”

Response: We have removed the word “potential” from line 10, and have included reference to the paper T. Kovachy, P. Asenbaum, C. Overstreet, C. A. Donnelly, S. M. Dickerson, A. Sugarbaker, J. M. Hogan and M. A. Kasevich, Quantum superposition at the half-metre scale,

Nature, 528, 530–533 (2015). Additionally, in line 143 we have added “As a comparison to an unconfined system, Ref.[*Nature*, 528, 530–533 (2015)] applies $90\hbar k$ of momentum separation between components of a cold atomic gas and achieves a spatial separation of 54 cm after allowing the components to drift apart for 1s.”

2) “Line 30-, “the natural localised state is a coherent state.” This could be phrased more understandably in terms of expressing that a coherent state is a displaced (in position, momentum, or some combination of both) quantum ground state. This makes more sense as a route to justifying it as natural localised quantum state and makes discussions of coherent superposition states more intuitive with regards to spatial or momentum displacements. It would also do well here to introduce the spatial/momentum tradeoff that the trapping provides. As opposed to the one way street that occurs in free expansion/time of flight type experiments in neutral atoms.”

Response: We have changed the sentence in line 30 to “The natural localized quantum state of a harmonic oscillator is its displaced ground state (coherent state) $|\text{ket}\{\alpha\}\rangle$ [cite{Glauber_1963_PR}], which is a Poissonian superposition of oscillator quanta with mean $|\alpha|^2$.” We have also added the clause “...and as energy oscillates between its kinetic ($\hat{p}^2/2m$) and potential ($m\omega^2 \hat{x}^2/2$) forms, the coherent state makes circles in phase space.” to line 34.

3) “Line 36- Could the authors please clarify why they’ve chosen to cite Ref 18 (Wineland Nobel Prize Review Mod Phys paper) regarding creating cat states in a massive particle instead of the original work by the last author C. Monroe*, D. M. Meekhof, B. E. King, D. J. Wineland A “Schrödinger Cat” Superposition State of an Atom Science 1996 Vol 272 pp 1131-1136. There is also the issue of the use of the singular particle vs. plural particles in regards to this statement as the work in question was using a single 9Be^+ ion. Using another/additional citation where multiple massive particles were in a cat state might also be appropriate here.”

Response: The reference in line 36 regarding cat states in massive particles has been changed to C. Monroe, et al., “A Schrödinger Cat” Superposition State of an Atom Science 1996 Vol 272 pp 1131-1136. The words “particles” and “fields” have been changed to “particle” and “field”.

4) “Lines 43-45 - Could be more clear, particularly given that Ref 21’s abstract explicitly states that they ‘experimentally observe the behaviour outside the Lamb-Dicke regime.’”

Response: We modified the statement to be “...where the motion is near or smaller than the wavelength...”, and added a new figure (now Fig. 1) explaining the relationship between the experiments that have made states around the Lamb-Dicke limit.

5) “Line 79- What is the wavelength spread of this laser pulse? Does it hit the P 3/2 state at all? What do you mean does not couple directly to any of the excited states? What is the excitation probability for driving a spontaneous decay? Its the wavelength spread, rather than the timing, which determines if the P1/2 and P3/2 state are excited. Alternatively stating something about the time-bandwidth product of the pulse would also indicate how Fourier transform limited its behaviour is.”

Response: The pulse is transform limited with ~ 100 GHz bandwidth, detuned from the $P_{1/2}$ state by 33 THz and the $P_{3/2}$ state by 67 THz. We expect the spontaneous emission probability per kick to be approximately 10^{-5} . We modified the text and provided an additional citation to previous work that describes this and the probability of driving spontaneous decay. We have also added the statement "...but is narrow enough not to resonantly excite any higher energy states; the center wavelength is detuned from the $^2P_{3/2}$ and $^2P_{1/2}$ levels by 67 THz and 33 THz respectively." to make clear that the $P_{3/2}$ state is not hit.

6) "Line 82- "arrive at the ion simultaneously" With regards to a 10 ps pulse. Are they within 0.1 ps of each other? 1 ps of each other? What is the tolerance here. The word "simultaneously" makes me nervous as a physicist when dealing with laser pulses. It would also be helpful to observe that the pulses are $\sim 1/1000$ the excited state lifetime of the P state lifetime, for those not familiar with the specifics of Yb+'s level structure."

Response: We have included an error estimate (" ± 70 fs") in the arrival time of the pulses. We appreciate the comment about lifetime, but feel that comparing the pulse duration to the excited state lifetime could confuse readers into thinking that we resonantly drive the excited states. Instead, we have added a statement about the linewidth of the excited states.

7) "Line 86- Clarify what sort of pulse the polarisation is creating. I understand its lin perp lin, but what is the B field orientation in the system. E.g. are these all driving Π transitions, Σ^+ , Σ^- , etc.? Please add a brief clarification for those not familiar with the details in Ref 26 regards how the state dependent force comes about (pulse ordering of absorb/emit with respect to state)."

Response: We have added the statement "...which are both orthogonal to an applied static magnetic field..." and added a magnetic field to the figure showing pulse configuration (now Fig. 2). We took care, using two paragraphs, to describe how equation (2) is made. Because this is a complicated process, we believe additional clarification should be sought in (previously) ref 26 for those curious.

8) "Figure 1b- the red vs. blue in the ground hyperfine splitting is barely visible. Thicken this up! Figure 1c- Are you intending to show the spin dependent kick also flips the spin state? Figure 1d. Where are the arrows in the lower half of the diagram to show the spin dependent kicks increasing the "orbital radius" Figure 1d,e. Would be good to remind people here that $\text{Im}(\alpha)$ is also momentum and $\text{Re}(\alpha)$ is position."

Response: We have thickened the red and blue lines in (previously) Fig. 1b. (Previously)

Fig. 1c- We have added the sentence "Each momentum transfer is accompanied by a spin flip." to the legend.

Fig. 1d- We have added arrows to the lower half of Fig. d.

Fig. 1d,e- We have changed the axes to be labeled with momentum and position.

9) "Figure 1 caption. Typo. "the kicks to occur at each half-period of oscillation of the ion in trha trap" "the kicks to occur at each half-period of oscillation of the ion in the trap.""

Response: We have fixed the typo, changing to “the kicks to occur at each half-period of oscillation of the ion in the trap.”

10) “Line 103-104- “A remarkable feature of this interaction is that it does not rely on confinement to the Lamb-Dicke regime”- Does this mean that it wasn’t necessary to cool to near the ground state in order to perform this experiment? Or does it just change the fidelity?”

Response: We have expanded this sentence to “A remarkable feature of this interaction is that it is blind to the Lamb-Dicke regime, where $\eta\sqrt{2n+1}\ll 1$ ” cite{Mizrahi_2013_APB}, with no reliance on a tightly confined initial and final state.” An additional comment to the reviewer: the form of the initial state does not affect the fidelity of a kick, however, running repeated experiments while ground state cooling slows the experiment down and allows for additional noise to affect the system (this is mentioned later in the paper). On the other hand, as mentioned in (new) line 154, for hotter initial states, the interference between the cat components occurs over a narrower range in phase space, requiring better trap stability when data is averaged over many runs.

11) “Line 111, For context What are the dimensions of your trap in the direction you're kicking”

Response: In the last sentence of this paragraph, the trap dimension is already stated.

12) “Line 115- by directions do you mean polarisation directions?”

Response: We have added the statement “...(by swapping the paths of propagation)...”.

13) “Line 154 “Multicomponent superposition states have not previously been created in the motion of atoms.”- This statement is not correct in the context of atom-interferometry and trapped neutral atom BEC type experiments. See “Optics and interferometry with atoms and molecules” Alexander D. Cronin, Jörg Schmiedmayer, and David E. Pritchard Rev. Mod. Phys. 81, 1051 (2009) For instance in Fig 2 dating back to 1930 shows atom diffraction peaks, thus a superposition of three states in the motion of atoms.”

Response: We think this is referring to (previous) line 45 instead of 154. We have changed this statement to “Multicomponent superposition states have not previously been created in the harmonic motion of trapped ions.”

14) “Figure 2 caption- Mention here that the period $T=1000$ ns to given context for the 62 ns to get $\alpha=1.2$ and 111 ns to get $\alpha=2.0$.”

Response: We have added the statement “($\omega/2\pi=1.0$ MHz)” in the 3a caption to remind the readers of the trap frequency. In the plots of 1b and 1c, it is directly inferred that we are using delay times T centered around one trap period, or $T=1000$ ns.

15) “Line 172- Please clarify what you mean by larger? Larger momentum difference, or larger spatially. Though these are linked, larger spatially also has to do with confinement and trap anharmonicity. E.g. is it larger α that is of interest, or larger enclosed area.”

Response: We mean a larger separation of coherent state components in position-momentum phase space. This is essentially characterized by the quadrature sum of the difference in position

with the different in momentum, properly scaled. We have modified the statement to “Multicomponent states with larger phase-space separations can be made in the same way; however, measuring...”

16) “Line 179- The authors might do well to speculate as to the applications of such large cat states to a more extended range of problems in quantum gravity or a Heisenberg’s microscope type problems.”

Response: We hesitate to speculate on connecting this work to gravitational experiments, as the mass of a single atom is likely too small to offer any advantages over more macroscopic media, even given the phase space separation in the experiment. However, we appreciate the suggestion, so we have added a further line in that paragraph. It now reads:

“In order to generate larger separations, the trap frequency could be substantially lowered: operating the current apparatus at $\omega/2\pi = 10$ kHz could produce cat state separations as large as $20 \mu\text{m}$. This would enhance the sensitivity interferometric measurements of rotation [30] or proximal electric field gradients. At such large separations, we can directly resolve the cat components with high resolution imaging techniques [31], allowing investigations in measurement backaction and ‘Heisenberg’s microscope’ type thought experiments [Ref: W. Heisenberg, Z. Phys. 43, 172 (1927)].”

17) “184-185- What are the consequences of this laser's rep rate not being actively stabilised. I’ll note its frequency drift is on the order of what I’m guessing is the the actively stabilised trap frequency drift in Ref 23. Again this depends on if its the same or a similar experimental apparatus.”

Response: We have added the statement “(or a 0.8 urad/min drift in θ at on trap period, which is insignificant for all data presented here).”

18) “222- What is the divergence of the focused Raman beams? Will this spread in angles also couple to the other directions? I would be concerned that this spread in angle is more important than the misalignment.”

Response: We have added the statement “The Raman beam waist is $\approx 3 \mu\text{m}$, with a divergence of about 2° ; the Rayleigh distance is $80 \mu\text{m}$, meaning that the ion would need to venture a considerable distance farther than it does during its excited oscillations in order to see an appreciable fraction of the divergence change in kick direction. Additionally, the axial component of a beam at its focus (with a waist of $3 \mu\text{m}$) is less than about 0.001% and is negligible.”

19) “225- Do you have an estimate of the trap anharmonic length scale(s). E.g. the ratio of the 2nd order quadratic confinement driving potential curvature term vs. the 3rd order cubic potential coefficient or 4th order quartic term. I’m assuming your electrodes are not precisely crafted conic section.”

Response: The trap potential scales as a polynomial in the ratio (x/d) , where x is position and $d=100\mu\text{m}$ is the distance to nearest electrode. By symmetry, we expect only even terms in the

polynomial, so to lowest order the potential is $(x/d)^2 [1 + C(x/d)^2]$, where the parameters C is of order unity. As stated on line 119, the criterion for weak anharmonicity is $x < d$, or equivalently $\alpha < d/x_0$, where x_0 is the zero-point width.

20) “Capitalisation of paper titles: Ref 6 – Raman Ref 7,13,18 – Schrödinger Ref 21- Lamb-Dicke”

Response: The capitalisation errors have been corrected.

22) “Typo: Ref 25 +Yb should be Yb+”

Response: The typo has been fixed.

23) “A definite weakness of the manuscript is the indirect readout of the more complex spatial superpositions, particularly in light of Ref 30, where the same group (though perhaps a different experimental apparatus?) achieved nanometer scale localisation of Yb+ ions. With a maximum spatial separation of 209 nm, I would have hoped some rough spatial information might be recoverable by deconvolution techniques.”

Response: The nanoscale resolution achieved in Ref. 30 is possible with integrations times in the order of hundreds of milliseconds for a different isotope ($^{174}\text{Yb}^+$) about three times brighter than the one used in this manuscript. In order to gain information about the components of the state, we would also want to scatter and collect light from the ion for a period of time much shorter than a trap period; this would require camera triggering on the order of few hundreds of nanoseconds. Additionally, each measurement collapses and disturbs the state, so this experiment would need to be repeated many times, likely over the course of many hours. Although these problems could be circumvented, due to technical hurdles and long-time stability issues, we were unable to capture such spatial distribution. We envision being able to achieve this spatial imaging of cat states in the future, however, we also believe that the data presented in this manuscript clearly shows the creation and verification of cat states.

Reviewer #3 had the following to say:

In their paper "Ultrafast create of large Schrodinger cats states of an atom" the authors present experiments generating and characterizing a variety of large spin-motion entangled states using a pulsed laser.

I think that this paper presents novel, interesting, results, and is well written. Once the authors address the below minor comments and questions I believe this work merits publication in Nature Communications.

With 5 comments:

1) “The phrase 'ultrafast' is used throughout the paper, yet the operations that create the large cat states are not that fast (tens of micro-seconds), due to repeated delays waiting for motional precession. I think the authors should explicitly mention the time-scales for creating the large

entangled states using the trap evolution method, as they do for faster method in the caption of figure 2.”

Response: We have added the sentence “These states are generated in times of $\sim(\Delta\alpha-0.4)\times 1250$ ns.” to the caption of Fig. 2c.

2) “Around line 148 the infidelity of the SDKs from trap frequency drift, with and without sub-doppler cooling, is discussed. It would be helpful for there to be some mention of the magnitude of frequency drift that would cause such an error, as well as the relative data collection time with and without sub-doppler cooling.”

Response: We have added a reference to Methods, where we added the statement “Ground state cooling increases data acquisition time by a factor of about 7 times. This would increase the acquisition time of a single, 10 point contrast lineshape from about 20 s to 140 s. While it is difficult to accurately assess the behavior of trap frequency noise on that time scale, trap frequency drifts on the order of kHz would cause the difference in fidelity seen between the revival lineshape of large cat states made from ground state cooled oscillations and Doppler cooled only states. Based on standard noise models, the 7 \times increase in acquisition time is likely the cause of this difference.”

3) “In Methods (d) the angle between the Raman delta-k and the desired mode is said to be < 0.5 deg, and it is said that this will set an upper limit on the 8 component cat contrast of 58(3)% - this seems to be quite a small uncertainty. If this is not false precision, the measured mode projections used to calculate this should be stated.”

Response: We have modified this statement to “The trap axes are rotated so that the Raman beam nominally couples only to a single mode. We estimate a misalignment of $< 0.5^\circ$, which entangles $< 0.8\%$ of the qubit population with other perpendicular modes of motion for each SDK. The alignment can be made to have a tighter tolerance, likely by at least a factor of 10 by using more sophisticated alignment methods.”

4) “It seems like it might be relevant to cite PRL 116 140402 (2016)”

Response: We have added a reference to PRL 116 140402 (2016).

5) “Typographics: *line 60: I assume this should be “ $w = w_x, w_y, w_z$ ” (not “ $w_x = w_y, w_z$ ”); * caption of fig. 1, (e): trha -> the; * line 220: suscpetibility -> susceptibility”

*Response: *Line 60 is written as intended: we are defining w to be $w_x=1$ MHz for the rest of the paper. We have fixed the other two typos.*

Additional changes to the revised manuscript:

We have added a figure (now Fig. 1).

We have added the abbreviation LDR

We have removed the punctuation from the title

We have moved any citations in the abstract into the Introduction.

We have added titles Results and Discussion to appropriate sections.

REVIEWERS' COMMENTS:

Reviewer #1 (Remarks to the Author):

The authors have satisfactorily replied to all of my comments and concerns. I think that the manuscript is now ready for publication in Nature communications.

Reviewer #2 (Remarks to the Author):

Only a few minor comments

Old Line 80- if the pulse is transform limited please also give the bandwidth for those who are not as familiar with ultrafast work.

Line 91, could you please state the bandwidth explicitly so its easy to compare against the detuning, hyperfine splitting, and natural line width.

In Fig 1b. why use ω_0 for the ground state HFS? In the previous version this was at least the more easily understandable ω_{hf} ? Also with the fine structure splitting in Hz, mixing and matching in radian frequency could cause confusion. Could you instead explicitly state the HFS splitting as 12.6 GHz. It would be good to also include the other relevant frequency widths, namely the pulse bandwidth and the excited state lifetime(s). Fig 1b should be able to quickly set the energy "landscape" of the experiment to the reader.

I concur with the authors that linewidth is more relevant for comparison than the excited state lifetime, as P isn't meant to be populated.

Old Line 86- I concur this is a complicated process. Present text is much improved for clarity to a non-expert/Ref 26 reader.

Old Line "154"- This was a typo on my part. Yes was line 45

Comment 23- I'm not surprised by this response and certainly consider it sufficient given the technical differences between the two experiments. I'm persuaded they're cat states, however that being said the indirect interferometric readout isn't as persuasive as an imaging result.

Reviewer #3 (Remarks to the Author):

The authors have addressed my comments and I am happy for the paper to be published in its current form.

Reviewer #1 had the following to say:

The authors have satisfactorily replied to all of my comments and concerns. I think that the manuscript is now ready for publication in Nature communications.

Reviewer #2 had the following to say:

Only a few minor comments

1) Old Line 80- if the pulse is transform limited please also give the bandwidth for those who are not as familiar with ultrafast work.

Response: We have added explicit statement of the bandwidth value.

2) Line 91, could you please state the bandwidth explicitly so it's easy to compare against the detuning, hyperfine splitting, and natural line width.

Response: See response to comment #1 (it is concerning the same issue as old line 80 was new line 91).

3) In Fig 1b. why use ω_0 for the ground state HFS? In the previous version this was at least the more easily understandable ω_{hf} ? Also with the fine structure splitting in Hz, mixing and matching in radian frequency could cause confusion. Could you instead explicitly state the HFS splitting as 12.6 GHz. It would be good to also include the other relevant frequency widths, namely the pulse bandwidth and the excited state lifetime(s). Fig 1b should be able to quickly set the energy "landscape" of the experiment to the reader.

Response: We have changed ω_0 to ω_{hf} in all instances throughout the manuscript. We have also explicitly state the 12.6 GHz hyperfine splitting in Fig. 1b. Finally, we have added all relevant splittings and linewidths to Fig. 1b.

4) I concur with the authors that linewidth is more relevant for comparison than the excited state lifetime, as P isn't meant to be populated.

5) Old Line 86- I concur this is a complicated process. Present text is much improved for clarity to a non-expert/Ref 26 reader.

6) Old Line "154"- This was a typo on my part. Yes was line 45

7) Comment 23- I'm not surprised by this response and certainly consider it sufficient given the technical differences between the two experiments. I'm persuaded they're cat states, however that being said the indirect interferometric readout isn't as persuasive as an imaging result.

Reviewer #3 had the following to say:

The authors have addressed my comments and I am happy for the paper to be published in its current form.